# Inferring hierarchies of latent features in calcium imaging data

**Luke Y. Prince**
University of Toronto Scarborough
luke.prince@utoronto.ca

**Blake A. Richards**
MILA
blake.richards@mcgill.ca

## Abstract

A key problem in neuroscience and life sciences more generally is that the data generation process is often best thought of as a hierarchy of dynamic systems. One example of this is in-vivo calcium imaging data, where observed calcium transients are driven by a combination of electro-chemical kinetics where hypothesized trajectories around manifolds determining the frequency of these transients. A recent approach using sequential variational auto-encoders demonstrated it was possible to learn the latent dynamic structure of reaching behaviour from spiking data modelled as a Poisson process. Here we extend this approach using a ladder method to infer the spiking events driving calcium transients along with the deeper latent dynamic system. We show strong performance of this approach on a benchmark synthetic dataset against a number of alternatives.

## 1 Introduction

In-vivo two-photon calcium imaging provides systems neuroscientists with the ability to observe the activity of hundreds of neurons simultaneously during behavioural experiments. Such high-dimensional data is ripe for techniques identifying low-dimensional latent factors driving neural dynamics. The most common methods, such as principal components analysis, ignore non-linearity and temporal dynamics in brain activity. Pandarinath et al. (2018) [1] developed a new technique using deep, recurrent, variational auto-encoders which they named Latent Factor Analysis via Dynamical Systems (LFADS). Using LFADS they found non-linear, dynamic latent variables describing high-dimensional activity in the motor cortex that can decode reaching behaviour with much higher fidelity than other methods. However, LFADS was designed for application to spiking data recorded from extracellular electrodes, not for two-photon calcium imaging data. Two-photon calcium imaging poses the additional problem of identifying latent spike trains in fluorescence traces. If we continue to model the frequency of events as being generated by a Poisson process, this can be seen as hierarchy of dynamic systems (Fig 1A), in which low dimensional dynamics generate spike train probabilities that drive fluctuations in biophysical dynamics of calcium activity (Fig 1B. Here we propose a method that extends LFADS to accommodate calcium activity using this hierarchical dynamic systems approach.

## 2 Model

The model is a variational ladder autoencoder (VLAE) [2] with recurrent neural networks (RNNs) that supports uncovering latent dynamic systems (Fig 1C). It can be seen as a unification of two recent applications of variational autoencoders (VAEs) in neuroscience: 1) Latent Factor Analysis for Dynamic Systems (LFADS) [1] and 2) DeepSpike, a VAE approach to inferring spike counts from calcium imaging data [3]. We choose the VLAE approach since it has been shown to learn disentangled hierarchical features, in contrast to stacked VAEs or ladder VAEs [2, 4].

Real Neurons and Hidden Units Workshop at the 33rd Conference on Neural Information Processing Systems (NeurIPS 2019), Vancouver, Canada.

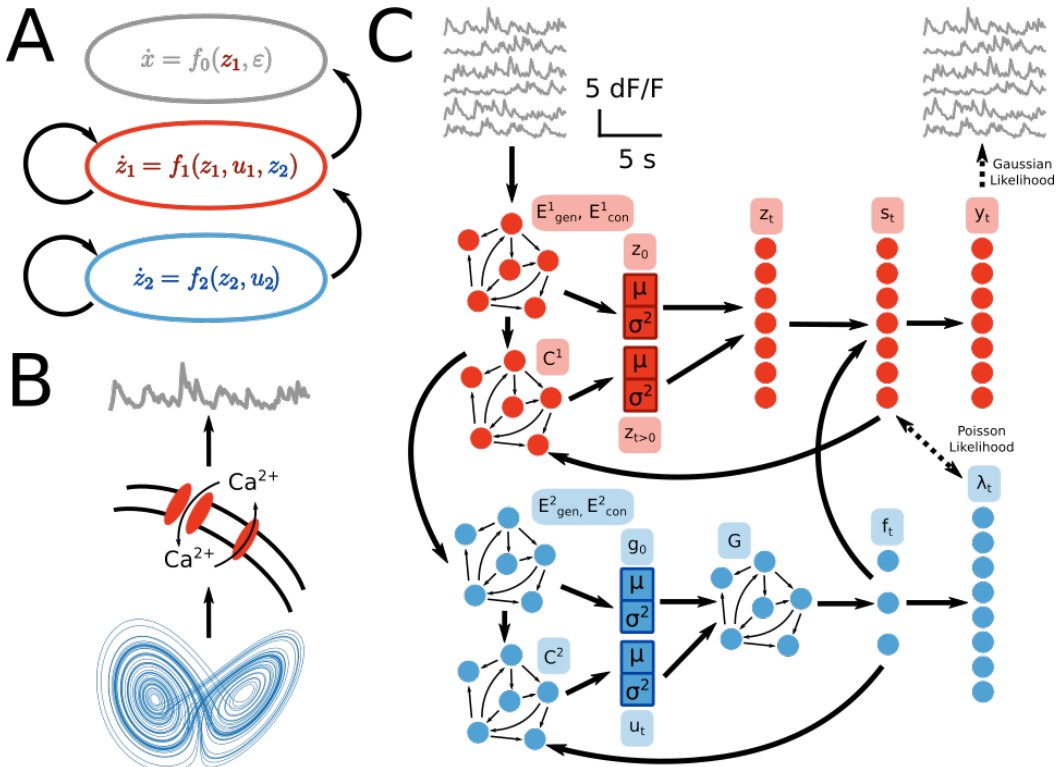

Figure 1: A) Hierarchy of dynamic systems, B) Schema of calcium and Lorenz dynamics, C) Schema of our hierarchical model

## 2.1 Generative Model

The inferred dynamic system underlying the frequency of calcium events in the data is identical to that of LFADS (Fig 1C, blue modules). The prior distribution of initial conditions $g_0$ and external inputs $u_t$ are modelled as Gaussian distributions $P(g_0) = \mathcal{N}(\mu_{g_0}, \sigma^2_{g_0})$, and $P(u_t) = \mathcal{N}(\mu_{u_t}, \sigma^2_{u_t})$. The underlying dynamic system $G(g_t, u_t)$ is modelled by a Gated Recurrent Unit (GRU) taking the initial hidden state $g_0$ and inputs $u_t$. Low dimensional factors $f_t$ are calculated as a linear transformation of the generator hidden state $f_t = W_{fac} g_t$. These factors are used to reconstruct the Poisson process intensity function with a fully connected layer and exponential non-linearity $\lambda_t = \exp(0.5(W_\lambda f_t + b_\lambda))$

Inferred spike counts $s_t$ are generated by sampling $z_t$ from Gaussian distributions $P(z_t) = \mathcal{N}(\mu_{z_t}, \sigma^2_{z_t})$ and projecting these through an affine transformation and non-linearity along with the factors from the deeper layer, i.e., $s_t = \Phi(W_s[f_t, z_t] + b_s)$, where $\Phi(x) = ReLU(\exp(x) - 1)$ (Figure 1C blue modules). We assume a simple model of calcium dynamics: $y_t = -y_t/\tau_y + \alpha_y s_t + \beta_y$ where the parameters $\tau_y, \alpha_y, \beta_y$ are measured from the data, however it is a topic for future research to fit the calcium dynamics simultaneously. In our synthetic data, these are valued at 0.4 s, 1, and 0 respectively. The value of $\tau_y$ is chosen as it is the known decay time constant of GCamP6, a commonly used calcium fluorescence indicator used in calcium imaging experiments.

## 2.2 Encoding Model

The variational posterior distributions $Q(z_t|x)$, $Q(g_0|x)$, $Q(u_t|x)$ are modelled as Gaussian distributions, with the mean and standard deviations parameterised by a stack of bidirectional GRUs, $E^1_{gen}$, $E^2_{gen}$, $E^1_{con}$, $E^2_{con}$. The final hidden states of $E^1_{gen}$ and $E^2_{gen}$ are mapped onto the parameters of $Q(z_0|x)$ and $Q(g_0|x)$ respectively with fully connected layers. The hidden states $E^1_{con}$ and $E^2_{con}$ are concatenated and passed as inputs to single direction GRUs $C^1$ and $C^2$. The hidden states of $C^1$

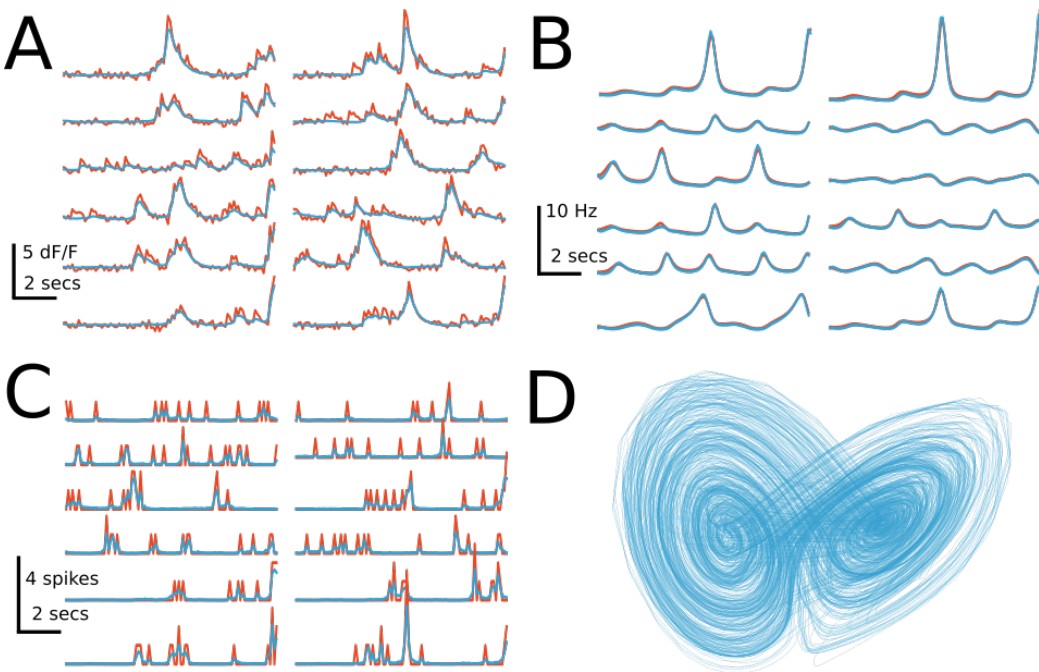

Figure 2: A) Example calcium fluorescence traces, B) Example Poisson intensity functions, C) Example spike trains, D) Inferred dynamics. Red: Ground-truth, Blue: Reconstructed

and $C^2$ are concatenated at each time step $t$ with $s_{t-1}$ and $f_{t-1}$. Subsequently these concatenated activities are mapped onto the parameters of $Q(z(t)|x)$ and $Q(u(t)|x)$ with fully connected layers.

### 2.3 Loss function and Training

One of the advantages of using VLAEs is that the evidence lower bound (ELBO) formulation is the same as for VAEs despite the hierarchical latent space [2]. As such, our cost function remains very similar to that of LFADS. The likelihood function $P(x_t|y_t)$ is modelled as a Gaussian distribution $x_t \sim \mathcal{N}(y_t, \sigma_y^2)$, where $\sigma_y^2$ is learned. Although $s_t$ is not discrete, $P(s_t|\lambda_t)$ is treated as an approximate Poisson process $s_t \sim Poisson(\lambda_t) = s_t^{\lambda_t} \exp(-\lambda_t)/\Gamma(s_t + 1)$.

Parameters of our model were optimized with ADAM, with an initial learning rate of 0.01, which decreased by a factor of 0.95 whenever plateaus in training error were detected. As in LFADS training, KL and L2 terms in the cost function were 'warmed up', i.e., had a scaling factor being 0 and 1 applied that gradually increased. Warmup for the deeper parameters (blue modules in Figure 1) was delayed until warmup for shallower parameters was completed (red modules in Figure 1).

## 3  Results

The model was tested on synthetic data with Lorenz dynamics embedded in the frequency of calcium fluorescence transients, as described in [5], where generated spikes were convolved with an exponential kernel with a time constant of 0.4 ms, and white noise added to the resulting traces. We measure the performance of the model in three ways: 1) uncovering the underlying Lorenz dynamics, 2) reconstructing the rates of calcium transients an inhomogenous Poisson intensity functions, 3) reconstructing the spike counts contributing to increases in the calcium fluorescence signal. The model was compared against a ground-truth where the spike counts are known, and LFADS is used to reconstruct the latent dynamics and intensity function, and against a model where spike counts are extracted using a deconvolution algorithm [6] before using LFADS to reconstruct the rates and intensity function (OASIS + LFADS). It was also tested against a model that used a 1-D convolution of the intensity function to reconstruct either the first two (Gaussian-LFADS) or four

Table 1: Comparison of model performance on synthetic Lorenz dataset. A hyphen indicates it is not possible to compare, as the model does not infer this variable. The top row is italicised as it is considered the upper limit on performance in this task with LFADS since there is no additional observation noise from fluorescence.

| Data | Model | End-to-end | Goodness-of-fit ($R^2$) | | | |
|------|-------|------------|--------|-------|--------|-------------|
| | | | Lorenz | Rates | Spikes | Fluorescence |
| *Spikes* | *LFADS* | - | *.978* | *.970* | - | - |
| Fluorescence | Oasis + LFADS | No | .924 | **.946** | **.950** | - |
| Fluorescence | Gaussian LFADS | Yes | .898 | .771 | - | .001 |
| Fluorescence | Edgeworth LFADS | Yes | .614 | .097 | - | .001 |
| Fluorescence | Ladder LFADS | Yes | **.962** | **.943** | .697 | **.850** |

(Edgeworth-LFADS) time-varying moments of fluorescence, as used previously in estimating the intensity functions of filtered Poisson processes in neuroscience [7].

Figure 2 shows examples of performance of our model in reconstructing the fluorescence traces (Fig 2A, Poisson intensity functions (Fig 2B), spikes (Fig 2C) and Lorenz dynamics (Fig 2D). Visually, the model provides very close fit to the fluorescence traces, intensity functions, and Lorenz dynamics. The model also captures spike-timing, although these spike trains appear smoothed. Table 1 compares the $R^2$ goodness-of-fit on reconstructing held-out validation data with ground-truth latent dynamic structure. Of all approaches, our model easily performs best at reconstructing fluorescence traces, and almost performs as well as LFADS in reconstructing the Lorenz dynamics. It is to be expected that LFADS performs better, since there is an additional source of observation noise in our synthetic dataset generating fluorescence transients from spikes. Notably, our model does not perform as well as the deconvolution method OASIS in reconstructing spike trains, however this does not impact the ability of our model to reconstruct the latent dynamics. In fact, constraining the reconstructed spikes by the latent dynamics may mitigate any errors in spike train reconstruction that occur by deconvolution, since the deconvolution algorithm may erroneously drop spikes during high rates, whereas our model should be less likely to do so. It will be necessary to assess this possibility further.

It should be noted that the deconvolution algorithm performs much better at reconstructing spike trains in our synthetic dataset than in real datasets where ground-truth spiking is known [8]. To our knowledge, there are no known dual recordings of population 2-photon calcium imaging with ground-truth electrophysiology in a subpopulation of neurons in-vivo during behaviour driven by hypothesized low-dimensional dynamics that we would be able to validate this with. Nevertheless, since the relationship between calcium dynamics and somatic spiking is highly non-linear, especially in dendrites, it remains to be seen how useful it is to faithfully reproduce unseen spike trains in calcium fluorescence activity.

## 4   Discussion

We present a hierarchical recurrent variational autoencoder model capable of reconstructing latent dynamics, latent spike trains, and calcium fluorescence traces in a benchmark synthetic dataset. Of the four methods tested, our model is the only one capable of reconstructing all three. Furthermore, our model performed best in reconstructing latent dynamics in our synthetic dataset We will need to assess our model on further synthetic benchmark data to assess the validity of our approach.

Since our model is trained end-to-end, it should be possible to extend to reconstructing raw 2-photon imaging videos, which could enable us to train models to uncover latent dynamics from arbitrarily shaped neuronal structures. This would of great use to neuroscientists who are largely restricted to techniques that extract fluorescence traces from regions of interest with somatic shapes, whereas the morphological diversity of dendrites is much greater.

We describe a use-case in neuroscience (2-photon calcium imaging data) for which this model may be very useful. However, we are keen to investigate the general case of hierarchical dynamical systems and their utility in uncovering structure in datasets outside this domain.

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
