# OpenReview forum: "Inferring hierarchies of latent features in calcium imaging data"
_NeurIPS.cc/2019/Workshop/Neuro_AI — Real Neurons & Hidden Units @ NeurIPS 2019 Poster_

### Official Review · AnonReviewer3 · 2019-09-23
**LFADS+fluorescence decay model effectively learns dynamics of spiking activity**

**Clarity:** 3

**Comment:**

Strengths:
The Ladder LFADS model does a good job of uncovering underlying dynamics and neural firing rates in simulated data. The combined approach outperforms a two-step approach where inference of latent dynamics follows a deconvolution step.

Areas for improvement:
Another useful benchmark would be replacing LFADS with a simple linear dynamical system. Though this would clearly fail in the case of Lorenz attractor dynamics, it seems like a natural comparison.

It will also be interesting to see how well this method works on real neural data from different brain regions. In regions like motor cortex, where dynamical systems models have been used for many years now, it seems this model will perform well. It is unclear how the model will perform, however, in sensory areas like visual cortex where activity is arguably more related to external inputs than internal dynamics.

I would also be curious to know how well the model works without using the ladder component; this seems like another natural comparison that could further motivate the modeling choice.

**Category:**

AI->Neuro

**Clarity Comment:**

The exposition was mostly clear, but I found it difficult to decipher the relationship between LFADS, stacked VAEs and ladder VAEs (as I have not heard of the last two before). Is the ladder feature necessary for this model to work, or does it merely improve the results? The motivation for using the VLAE approach is that it learns disentangled hierarchical features, but again it's not clear to me exactly how that is relevant here. Is it because the latent dynamics need to be disentangled from the calcium dynamics? A few clarifying sentences in the introduction of section 2 could go a long way to clearing up these ambiguities for me.

**Evaluation:**

3: Good

**Importance:**

4: Very important

**Importance Comment:**

Calcium imaging represents an important technological step forwards in our ability to record large populations of neurons. However, the increase in spatial resolution comes with a decrease in temporal resolution. Developing new algorithms for inferring the underlying neural activity at timescales shorter than fluorescence decay dynamics is crucial to taking full advantage of this data and the scientific insights it can lead to.

**Intersection:**

4: High

**Intersection Comment:**

The Ladder LFADS model is a combination of several recent neural network architectures that address an existing neuroscience problem in a new way.

**Rigor Comment:**

The proposed algorithm is a rigorous model-based approach to the problem.

**Technical Rigor:**

4: Very convincing

---

### Official Review · AnonReviewer1 · 2019-09-25
**Applications of VAEs to synthetic calcium imaging traces**

**Clarity:** 4

**Comment:**

This paper combines two previously published unsupervised VAE-based models and adapts them to infer the underlying dynamics of a synthetic calcium imaging dataset. While the results look excellent on the synthetic data, it's difficult to evaluate how great this method would be without seeing its performance on real traces.

**Category:**

AI->Neuro

**Clarity Comment:**

The paper is well-composed for the most part, I have some minor comments:
- Line 17: period missing in "brain activity Pandarinath"
- Lines 34-35: "We choose the VLAE approach ... in contrast to stacked VAEs or ladder VAEs". Is VLAE not a ladder VAE?
- Line 52: Reference for GCAMP6 time constant
- Line 65: open bracket )

**Evaluation:**

3: Good

**Importance:**

5: Astounding importance

**Importance Comment:**

Calcium imaging allows the simultaneous visualization of activity from thousands of neurons. Recordings from wider fields can lead to a lower SNR. Developing unsupervised methods that can reveal underlying dynamics or otherwise denoise the data is of critical importance for neuroscience.

**Intersection:**

4: High

**Intersection Comment:**

This paper is an example of applying AI (previously published unsupervised variational autoencoders) to the analysis of calcium imaging traces, a common neuroscience recording technique.

**Rigor Comment:**

The results in this paper look promising, but they come from (I believe) entirely synthetic data.
- Has the model been applied to real traces, and is there a way to reliability evaluate the quality of the output (spike train inference, underlying dynamics)?
- Line 75: How much white noise was added to the synthetic traces? How robust was the model to noise?
- Line 75: When generating the data, how does adding some noise to the time-constant affect the model?
- Line 69: How was the parameter for L2 regularization determined?

**Technical Rigor:**

3: Convincing

---

### Official Review · AnonReviewer2 · 2019-09-27
**Simple modification of well known model; more work is needed to pinpoint the advantages.**

**Clarity:** 4

**Category:**

AI->Neuro

**Clarity Comment:**

Well written and well presented.

**Evaluation:**

3: Good

**Importance:**

2: Marginally important

**Importance Comment:**

Estimating the dynamics of the latents while incorporating calcium dynamics is an important consideration. Estimating the calcium kernel with the dynamics is an interesting challenge. However, here the kernel is known by the authors, and it seems like they essentially stick on a known kernel at the output of LFADS. Moreover, given that the authors use relatively clean synthetic data, and the performance gains are minimal at best, it remains to be seen whether this approach has any advantages.

**Intersection:**

4: High

**Intersection Comment:**

Augmenting a machine learning model to estimate dynamics in neural data, although only synthetic at this point.

**Rigor Comment:**

The approach seems principled and technically sound. I commend the authors on their sincere and well-performed benchmarking. The actual results are just not much better than the stepwise deconvolution + LFADS, which is unfortunate. The authors could have added more noise to their dynamics / poisson observations, in order to clarify the regimes in which their approach may work better than the stepwise approach. The method has promise, though, and it would be interesting to see what it looks like on real data.



**Technical Rigor:**

4: Very convincing

---

### Decision · Program_Chairs · 2019-10-02

Accept (Poster)